# Plasma biomarkers of the amyloid pathway are associated with geographic atrophy secondary to age-related macular degeneration

**Kameran Lashkari**[1¤a]*, **Gianna C. Teague**[1], **Ursula Beattie**[1], **Joanna Betts**[2¤b], **Sanjay Kumar**[2¤c], **Megan M. McLaughlin**[2¤d], **Francisco J. López**[2¤e]

**1** Schepens Eye Research Institute, Mass Eye & Ear, Harvard Medical School, Boston, Massachusetts, United States of America, **2** Alternative Discovery & Development, GlaxoSmithKline, King of Prussia, Pennsylvania, United States of America

¤a Current address: Department of Bioengineering, University of Massachusetts- Dartmouth, Dartmouth, Massachusetts, United States of America
¤b Current address: Computational Biology, GlaxoSmithKline, Stevenage, United Kingdom
¤c Current address: Novel Human Genetics Research Unit, GlaxoSmithKline, King of Prussia, Pennsylvania, United States of America
¤d Current address: Development, GlaxoSmithKline, King of Prussia, Pennsylvania, United States of America
¤e Current address: Clinical Development Ophthalmology, Allergan an Abbvie Company, Irvine, California, United States of America
* klashkari@umassd.edu

**Data Availability Statement:** Data for this study are available at dx.doi.org/10.17504/protocols.io.bhwuj7ew.

## Abstract

Geographic atrophy (GA) is an advanced form of dry age-related macular degeneration (AMD), in which local inflammation and hyperactivity of the complement pathway have been implicated in its pathophysiology. This study explores whether any surrogate biomarkers are specifically associated with GA. Plasma from subjects with GA, intermediate dry AMD and non-AMD control were evaluated in 2 cohorts. Cohort 1 was assayed in a 320-analyte Luminex library. Statistical analysis was performed using non-parametric and parametric methods (Kruskal-Wallis, principal component analysis, partial least squares and multivariate analysis of variance (MANOVA) and univariate ANCOVAs). Bioinformatic analysis was conducted and identified connections to the amyloid pathway. Statistically significant biomarkers identified in Cohort 1 were then re-evaluated in Cohort 2 using individual ELISA and multiplexing. Of 320 analytes in Cohort 1, 273 were rendered measurable, of which 56 were identified as changing. Among these markers, 40 were identified in univariate ANCOVAs. Serum amyloid precursor protein (sAPP) was analyzed by a separate ELISA and included in further analyses. The 40 biomarkers, sAPP and amyloid-β (Aβ) (1–42) (included for comparison) were evaluated in Cohort 2. This resulted in 11 statistically significant biomarkers, including sAPP and Aβ(1–40), but not Aβ(1–42). Other biomarkers identified included serum proteases- tissue plasminogen activator, tumor-associated trypsinogen inhibitor, matrix metalloproteinases 7 and 9, and non-proteases- insulin-like growth factor binding protein 6, AXL receptor tyrosine kinase, omentin, pentraxin-3 and osteopontin. Findings suggest that there is a preferential processing of APP to Aβ(1–40) over Aβ(1–42), and

**Funding:** This work was supported by Grant #4100112, GlaxoSmithKline, King of Prussia, PA, USA (https://us.gsk.com) awarded to KL. Coauthors, JB, SK, MMcLM and FJL received funding from GlaxoSmithKline in the form of salaries. The funders of this study had no role in data collection and analysis or the decision to publish. GlaxoSmithKline authors contributed to study design, the interpretation of results and preparation of the manuscript.

**Competing interests:** On behalf of all authors, we have no competing interests in this study. This work was supported by Grant #4100112, GlaxoSmithKline, King of Prussia, PA, USA (https://us.gsk.com) awarded to KL. Coauthors, JB, SK, MMcLM and FJL received funding from GlaxoSmithKline in the form of salaries. The commercial affiliations for this work do not alter our adherence of PLOS ONE policies on sharing data and materials. There are no products in development or marketed products to declare.

a potential role for the carboxylase activity of the γ-secretase protein, which preferentially splices sAPPβ to Aβ(1–40). Other markers are associated with the breakdown and remodeling of the extracellular matrix, and loss of homeostasis, possibly within the photoreceptor-retinal pigment epithelium-choriocapillaris complex. These data suggest novel disease pathways associated with GA pathogenesis and could provide potential novel targets for treatment of GA.

## Introduction

Geographic Atrophy (GA), the advanced form of dry age-related macular degeneration (AMD), is characterized by progressive disorganization and subsequent loss of retinal pigment epithelium (RPE), photoreceptor (PR) and choriocapillaris (CC) layers in the macula leading to permanent vision loss [1]. Loss of visual function is related to the encroachment of initially spared fovea by peripheral GA lesions and their ultimate expansion into the fovea. Visual deficits are common in patients who have GA lesions around the foveal center; these patients are unable to read even though they have normal visual acuity as assessed by the Snellen chart. During reading saccades, words fall into the blind spots associated with neighboring GA lesions making it difficult for the patient to read [2]. The incidence of GA parallels that of neovascular AMD with rates of 1.9 and 1.8 per 1000 Caucasian Americans, respectively [3]. Examining the prevalence of AMD, Wong et al. (2014) have projected that 196 million people will be affected by AMD in 2020, increasing to 288 million by 2040 [4]. Overall, approximately half of the worldwide AMD population will have advanced AMD, including GA, with total prevalence of over 4–6% [4].

Although the exact pathophysiology of GA is not known, associations with age, smoking and body mass index have been described [5]. In earlier stages of dry AMD, there is deposition of drusen, as well as lipids and other cellular debris in the Bruch's membrane. Inflammatory proteins and lipids associated with these deposits are believed to trigger immune responses primarily mediated through activation of the alternative complement pathway [6–8]. Anatomically, these changes result in alterations in RPE metabolism leading to the "stressed" RPE phenotype and eventually the death/loss of RPE cells [9]. In addition to drusen deposition as a risk factor to progression of GA, reticular pseudodrusen (RPD) are also associated with GA [10]. RPD are easily visualized on near infrared (NIR) and OCT imaging and are characterized by accumulation of triangular-shaped material in the subretinal space and are associated with RPE phenotypic changes [11]. Although there are many commonalities between drusen and RPD, RPD has higher concentrations of unesterified cholesterol, vitronectin and PR constituents [12].

Genetic susceptibility markers for AMD and GA have been described, and several are associated with the complement system in general and the alternative complement pathway in particular. These include variants of complement factor (CF) H, CFI, CFB, C3, C4A/B and C9 among others [13–19]. Based on their genetic sequences, these variants are theoretically predicted to lead to heightened activation of the alternative complement pathway. To this end, Reynolds et al. (2009) [20] have shown elevated levels of C3a, Ba, Bb, and C5a in plasma of subjects with GA. Translating this information into therapy has been difficult, as recent phase 3 clinical trials that targeted components of the alternative complement pathway failed [21]. Lampalizumab, an antibody that targets CFD and hence reduces alternative complement cascade activity, also failed to reach its clinical endpoint in Phase 3 clinical trials even though it showed promising results in the phase 2 (MAHALO) study [22, 23]. However, more general

inhibition of complement-mediated innate immunity via blockade of C3 has shown promising results in Phase 2 studies. APL-2, a derivative of compstatin, has recently been shown to reduce GA progression rate by approximately 30% in the FILLY study [24]. Based on these promising results, two Phase 3 studies (DERBY and OAKS) are aimed at confirming that blockade of complement-mediated innate immunity can reduce progression rate of GA (ClinicalTrial.gov identifiers, NCT03525600 and NCT03525613; see https://gastudy.com for additional information). More recently, a phase 2 study that evaluated an anti-complement C5 aptamer (Zimura®; Iveric Bio Inc.) has also shown promising data overall, suggesting that blockade of complement activation may be a plausible approach to reducing progression rate in this devastating disease (see https://investors.ivericbio.com/events/event-details/zimurar-rd-symposium for details).

Another pro-inflammatory pathway that may also interact with the alternative complement pathway, and which has not been fully characterized in GA, is the β-amyloid pathway. It has been shown that amyloid-β (Aβ) is a constituent of drusen [25] and contributes to induction and maintenance of a local inflammatory state [26–28]. Aβ may also induce toxicity at the PR level by impacting their ribosomal machinery and oxidative phosphorylation by increasing phosphorylation of tau protein and dysregulating glycogen synthase kinase-3 beta (GSK3β) [29]. We have recently shown that Aβ can inhibit the activity of CFI, one of the key breaks in the alternative complement pathway leading to a shift in overproduction of C3b (active pathway) in lieu of production of iC3b which as a consequence inactivates the pathway [30]. Increased C3b leads to the eventual production of C5b678(9) membrane attack complex via the alternative complement pathway. However, a clinical study attempting to block the action of Aβ via systemic administration of an anti-Aβ antibody (GSK933776) failed to reduce GA progression rate [31], probably because of the inability of systemic administration to achieve sufficiently high concentrations of the antibody at the level of the RPE/Bruch's membrane interphase [30].

Aβ is derived from cleavage of its precursor, the amyloid precursor protein (APP). APP is cleaved by α- or β-secretase resulting in soluble (s)APPα or sAPPβ. Cleavage of APPα and APPβ by γ-secretase results in production of P3 and tail peptide (ICD) or Aβ and ICD, respectively. At least 20 species of Aβ polypeptides have been identified of which Aβ(1–40) and Aβ (1–42) are predominant. Aβ molecules are present as monomeric, oligomeric and fibrillar conformations with oligomeric driving pathologic processes [32]. The Aβ pathway may also indirectly influence heightened complement activation of the alternative complement pathway [26, 30, 33]. APP also undergoes extensive processing, and mutations in APP are associated with familial Alzheimer's disease (AD) [34]. Cleavage of APP by α-secretase resulting in production of sAPPα and p3 (by γ-secretase), is involved in the non-amyloidogenic pathway. This pathway supports the health and proliferation of resident stem cells [35]. Alternatively, cleavage of APP by β-secretase results in amyloidogenic sAPPβ and downstream production of Aβs by γ-secretase. Aβ moieties are highly neurotoxic and abundant in senile plaques in AD [36]. Other than its presence in drusen deposits associated with AMD, the role of the Aβ pathway has not been thoroughly investigated in GA.

A plethora of serum and plasma biomarkers have been associated with AMD, but not specifically with GA. These include interferon gamma-inducible protein 10 (IP-10), eotaxin, carboxyethypyrrole, homocysteine, C-reactive protein, and microRNAs [37–39]. Inflammasome activation pathways have also been implicated in the pathogenesis of GA. Activation of the NLRP3 inflammasome pathway leads to caspase-I mediated production of IL-1β and IL-18 leading to activation of innate and adaptive immunity [40]. However, the direct connection between inflammasome activation and RPE cell death has not been defined [5].

In this study, we used a large multiplexing library (Myriad RBM discoveryMAP) to evaluate biomarker levels in a cohort of non-AMD control subjects and those with intermediate dry

AMD (AREDS 3 stage [41]) and GA. We then repeated the results in a second cohort using different methodology. Based on these results, we have identified biomarkers that are associated with GA and interestingly, intimately involved with the Aβ pathway. Identifying plasma biomarkers specific to GA would potentially hasten our understanding of disease processes, identify new therapeutic targets and may also help develop non-imaging prognostic endpoints.

## Materials & methods

### Study subjects

Plasma levels of biomarkers were analyzed in 180 subjects divided into 2 cohorts. Cohort 1 was comprised of 94 subjects (non-AMD control, n = 33; AREDS 3 [41], n = 24; GA, n = 37) and Cohort 2 was comprised of 86 subjects (non-AMD control, n = 29; AREDS 3 [41], n = 22; GA, n = 35). Subjects were recruited from 2 sources, the observation phase (pre-treatment) of the GSK Phase 2 multi-center trial of GSK933776 in adult patients with geographic atrophy (GA) secondary to AMD (ClinTrials.gov identifier NCT01342926; GSK study 114341), and from the practice of one of the authors (KL). Subjects were staged as non-AMD control (AREDS stage 0), intermediate stage dry AMD (AREDS stage 3), and advanced stage dry AMD—GA (AREDS 4) [41]. Subjects were recruited using a rigorous inclusion and exclusion criteria including: > 55 years of age, no major systemic diseases including diabetes, cancer, collagen vascular or auto-immune disease, no cognitive changes or prior diagnosis of CNS disease including AD, no recent general surgical or ocular procedures (<90 days), no previous treatments with biologics, chemotherapy or anti-inflammatory therapy, no significant ocular disease including glaucoma or vascular disease. Subjects in the GSK study group had undergone a baseline brain imaging study to rule out brain disease including AD [31]. Remaining subjects had a rigorous review of their medical record to assure that there were no cognitive conditions including AD.

Plasma collection procedures from each source were approved under separate Institutional Review Boards (IRBs), and according to the Declaration of Helsinki. A written informed consent was obtained from each subject. IRB and Ethics Committees included: Sterling IRB, Atlanta, GA, USA; Mass Eye & Ear IRB, Boston, MA USA; Western Institutional Review Board, Puyallup, Washington, USA; University of Utah IRB, Salt Lake City, Utah, USA; The Human Research Program, The Committee on Human Research, University of California San Francisco, San Francisco, CA, USA; The Methodist Hospital Research Institute, Institutional Review Board, Houston, TX, USA; University of Pennsylvania, Office of Regulatory Affairs, IRB Committee Number 5, Philadelphia, PA, USA; The Johns Hopkins Medicine IRB, Baltimore, MD, USA; Henry Ford Health System Research Administration IRB, Detroit, MI, USA; The University of Texas Medical Branch at Galveston IRB, Galveston, TX, USA; The University of California San Diego, Human Research Protection Program, La Jolla, CA, USA; University of California, Irvine, Office of Research Administration, Irvine, CA, USA; Health Sciences IRB, University of South California Health Sciences Campus, Los Angeles, CA, USA; Tufts Health Sciences Campus IRB, Boston, MA, USA; New York Eye & Ear Infirmary IRB, New York, NY, USA; University of Miami Human Subjects Research Office, Miami, FL, USA; University of Virginia, IRB for Health Sciences Research, Charlottesville, VA, USA; University of Kansas Medical Center, Human Subjects Committee, Kansas City, KS, USA; Institutional Review Board Services, Aurora, Ontario, Canada.

### Plasma collection

Plasma samples were collected in 8 mL citrate buffer plasma collection tubes (BD Biosciences, CPT, or equivalent), promptly centrifuged according to the manufacturer's instructions and

the plasma component was aliquoted and stored in -80˚C. In the subgroup of subjects recruited by GSK, all subjects had been previously diagnosed to have GA associated with AMD, using standard clinical techniques and confirmed by The Wisconsin Reading Center according to the established protocol. In the second subgroup (KL patients), subjects were evaluated with slit lamp biomicroscopy, fundus imaging (Cannon Digital Fundus Camera), OCT and fundus autofluorescence imaging (both, Spectralis, Heidelberg System, Heidelberg, Germany). Subjects were then staged for AMD using the AREDS classification, based on similar classification principles as used in the multi-center trials [41].

## Evaluation of biomarkers in Cohort 1

**Detection of 320 analytes by quantitative immunoassay.** Plasma samples collected from Cohort 1 were analyzed for 320 biomarkers using the Luminex xMAP technology (Discovery-MAP v. 3.3, Myriad RBM, Austin, TX). This assay methodology uses a microsphere-based multiplexing assay based on Luminex technology and follows the CLIA guidelines as previously described [42]. All study samples were analyzed in duplicate. Samples were arranged in an alternating manner (see Fig 1A, top right panel) with pooled quality control (QC) samples distributed throughout the plates. QCs included wells loaded with plasma collected from similar subjects that had been pooled to achieve the same concentration of markers. QCs were placed in several wells in each 96-well plate and the analysts were masked to their positions. Analytes that were measured below the lower limit of quantification (LLOQ) were rejected and no longer analyzed.

**Detection of sAPP by separate ELISA.** One additional analyte, sAPP, which was not included in the 320-analyte library, was analyzed in Cohort 1 by a separate highly sensitive ELISA (R&D Systems) according the manufacturer's instructions. Procedure for performing ELISA has been previously published [30].

**Procedure for Luminex assay.** The procedure for performing the Luminex assay has been previously published [43]. Antibody-coupled microsphere sets were suspended in assay buffer to a dilution of 50 microspheres per microliter. 50 μL of microsphere mixture was pipetted into 96-well plates. 3 mL of plasma samples were thawed, vortexed and diluted at least 1:5 (for most analytes) in buffer. For background wells, 50 μL of assay buffer was used; 50 μL of either study plasma, QC pooled plasma or standard were gently pipetted up and down into wells. Plates were covered and placed on a plate shaker at 800 rpm for 30 min. at room temperature (RT). Plates were placed on a magnetic separator for 60 sec. Wells were then washed twice by pipetting 100 μL of assay buffer, manually inverting the plate and aspirating the assay buffer. Plates were removed from the magnetic separator and microspheres were resuspended in 50 μL of assay buffer by gentle pipetting; 50 μL of biotinylated detection antibody (4 ug/mL) was added to each well and mixed by pipetting. Plates were placed on a plate shaker at 800 rpm for 30 min. at RT and then placed on the magnetic separator to allow for separation for 60 sec. The supernatants were gently aspirated, and wells were washed twice by adding 100 μL of assay buffer and aspirating the supernatant. Plates were removed from the magnetic separator and microspheres were resuspended in 50 μL of assay buffer. SAPE reporter (ThermoFisher, Waltham, MA; 4 μg/mL) was added to each well and mixed by pipetting up and down. Plates were covered and incubated on a shaker (800 rpm) at RT for 30 min. and placed on magnetic separator for 60 sec. Supernatant from each well were aspirated and plates were rewashed with 100 μL of assay buffer (x2). Plates were finally removed from the magnetic separator and microspheres were resuspended in 100 μL of assay buffer. Samples were then analyzed on the Luminex analyzer according to the system manual.

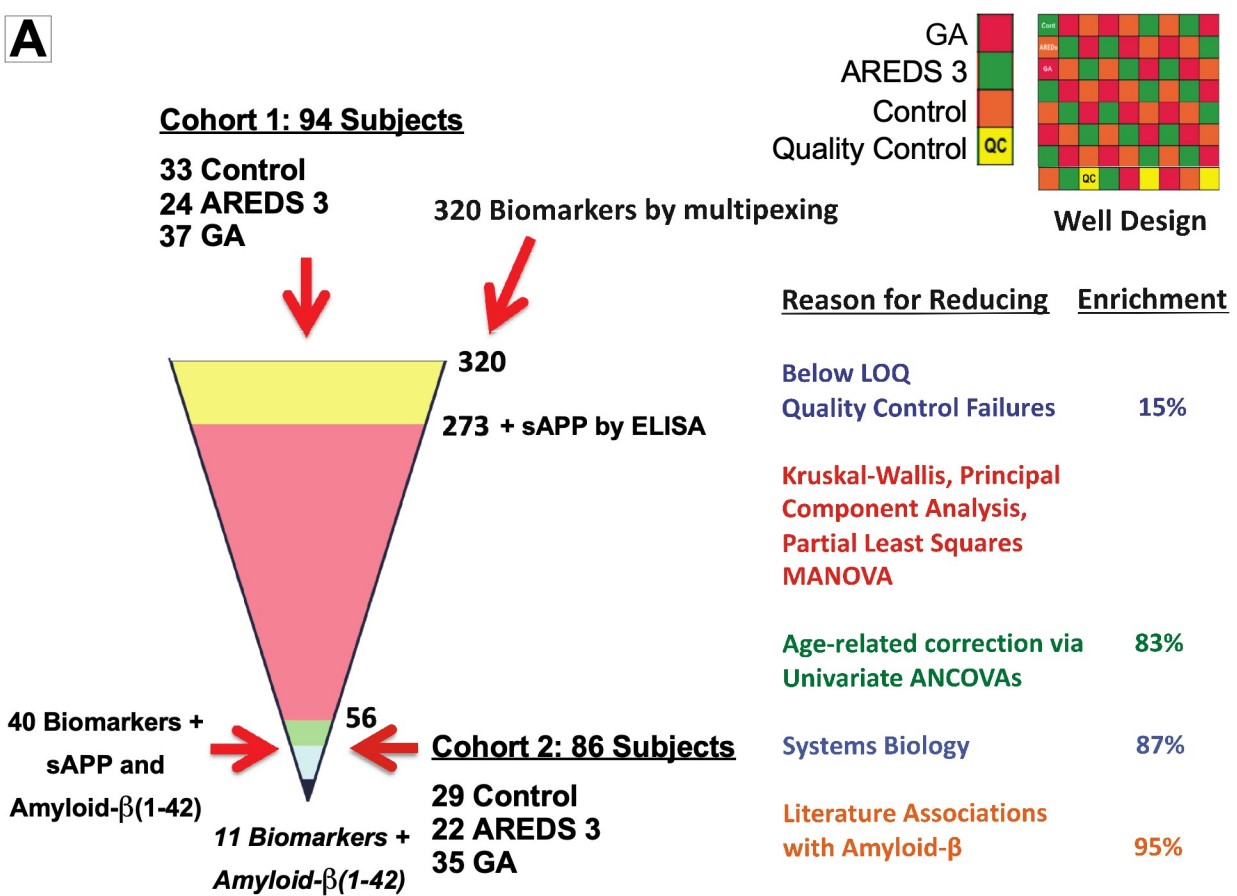

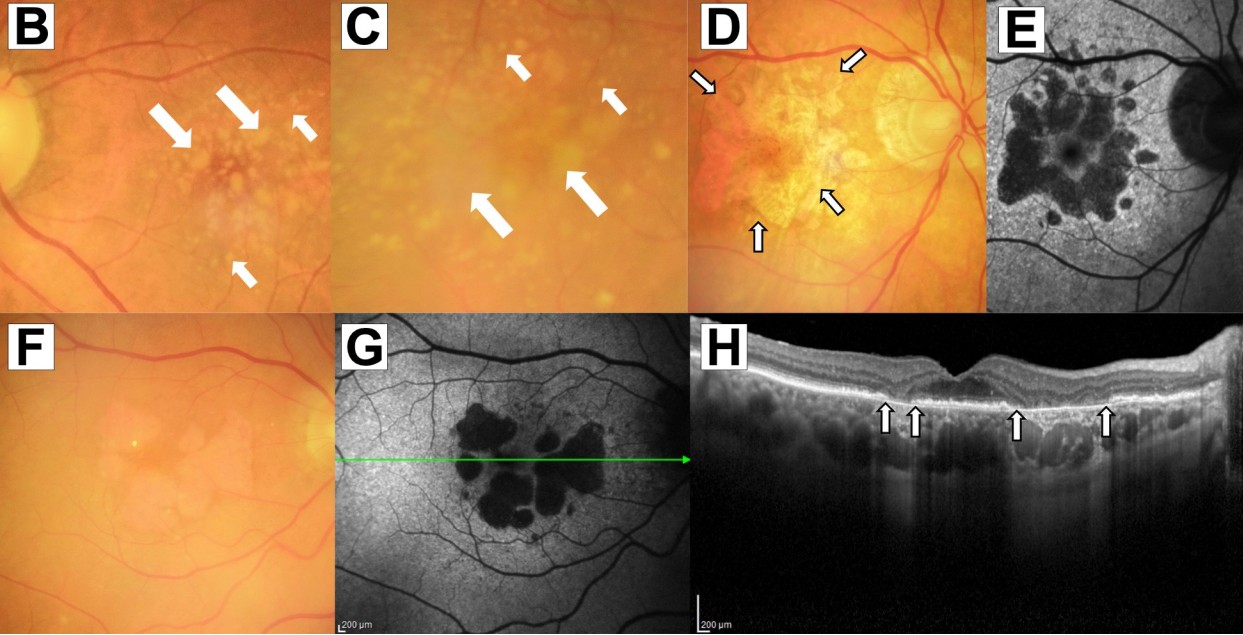

**Fig 1. Stepwise statistical approach to identifying reproducible biomarkers associated with GA and imaging of intermediate dry AMD and Geographic Atrophy (GA).** (A) 320 biomarker multiplexing was performed on 94 plasma samples. From 320 markers, 15% had below lower limit of quantification/quality control (QC) failures and were removed with 273 analytes being of sufficient quality for analysis. Soluble amyloid precursor protein (sAPP) was analyzed by a separate ELISA. Data from these biomarkers were evaluated by non-parametric and parametric analyses including Kruskal-Wallis, principal component analysis, partial least square, and MANOVA. Data were then corrected for age using

ANCOVA resulting in 83% enrichment with 56 biomarkers remaining (see S2 Table, highlighted in yellow). Data were Box-Cox transformed. Multiple comparisons were conducted using the Tukey test. Of 56 markers, 40 reached statistical significance; sAPP also reached statistical significance in a separate analysis. These 40 factors plus sAPP and amyloid-β(1–42) were selected for confirmation in another independent set of samples (Cohort 2) using ELISA and multiplexing. Data generated from ELISA of these biomarkers were subjected to similar statistical approaches and 11 were identified as significant in Cohort 2. (**B, C**) Color fundus images from intermediate dry age-related macular degeneration (AMD) (AREDS stage 3) show scattered and confluent soft (fluffy yellowish deposits, large arrows) and hard (well-demarcated deposits, small arrows) in the macula. (**D, F**) Central GA defined as loss of outer retina, retinal pigment epithelium (RPE) and choriocapillaris (CC) (outlines shown in small arrows). (**E, G**) Corresponding fundus autofluorescence images are shown. Areas of hypo-autofluorescence correspond to areas of GA, representing loss of RPE autofluorescence in response to blue light. (**H**) Optical coherence tomography cross-section image of the fovea from (**F, G**; scanned at green line) shows loss of outer retinal, RPE and CC (between arrows).

## Evaluation of biomarkers in Cohort 2

Three groups of markers (totaling 42) were identified in Cohort 1 and re-evaluated in Cohort 2 using either individual ELISA assays or multiplexing (Bio-Rad or R&D Systems). Sources of assay and their glossary are shown in S1 Table. Except for the ELISA for sAPP, these technologies are different from the Luminex xMAP assays used for Cohort 1 samples (see Fig 1A). These 42 markers included the following: (1) 40 markers were identified as changing using the Luminex xMAP assays; (2) one marker, Aβ(1–42), was identified as non-changing and added for comparison with Aβ(1–40); (3) one marker sAPP (related to the amyloid pathway and not included in the original 320-analyte library) was detected by a separate ELISA and identified as changing. These 42 analytes were then re-measured in Cohort 2. Regarding evaluation of the amyloid pathway, Cohort 2 samples were subjected to highly sensitive ELISAs for detection of sAPP (R&D systems), Aβ(1–40) and Aβ(1–42) (both from IBL International). Methodologies for performing ELISAs and multiplexing (Bio-Rad/R&D Systems) have been previously described and were performed according to the manufacturers' recommendations [30, 44].

## Bioinformatic network analysis

Genes corresponding to the proteins identified as differentially regulated in the GA group from Cohort 1 were evaluated using the Ingenuity Pathway Analysis (IPA) tool (Qiagen Ingenuity, http://www.ingenuity.com). A network algorithm approach was used to connect the genes through direct or indirect (transcriptional) relationships.

## Statistical analysis

Data from Cohort 1 were evaluated for precision using the QC samples included in the run for cohort. Among the 320 assays conducted, 273 markers were evaluated for further analyses. 47 analytes were discarded either because they lacked variability (e.g., most samples had values below LLOQ) or they failed QCs for the different assays. Kruskal-Wallis tests and principal component analysis (PCA) followed by partial least squares (PLS) coupled to the jack-knife statistical procedure were applied to all the 273 markers. These analyses yielded approximately 102 potentially significant markers. These 102 were then submitted to MANOVA after Box-Cox transformations to improve statistical properties of the data when appropriate [45, 46]. Individual P values from these evaluations were corrected for multiplicity using the Benjamini-Yekuteli [47] correction; to be inclusive, analytes with a corrected P value <0.1 were selected for further analysis. This final analysis resulted in 56 markers (Fig 1A; S2 Table, highlighted in yellow). These 56 markers were then subjected to univariate ANCOVAs after Box-Cox transformations as described above for MANOVA using age as a covariate. In the ANCOVA model, the fixed effect of disease state was coded as an ordered factor, which allowed for evaluation of linear trends. Multiple comparisons after ANCOVA were conducted using the Dunnett's multiplicity correction for comparisons versus a control. In this analysis

14 analytes were dropped because they exhibited either (1) significant differences between GA and AREDS 3, but not between GA and non-AMD control, or (2) significant differences between AREDS 3 and non-AMD control but not between AREDS 3 and GA or had >30% of their values being below the LLOQ. This resulted in 40 markers that were statistically significant (see S3 Table). As mentioned previously, sAPP detected separately by ELISA, was included in this data set and this analyte also showed statistically significant changes. Aβ(1–42), a statistically non-significant analyte was included for comparison with Aβ(1–40). Collectively, the total the number of markers were 42. These 42 markers were then re-evaluated in Cohort 2 for confirmation using different methodologies (see Fig 1A). Analyses in Cohort 2 were conducted using univariate ANCOVAs after Box-Cox transformations and multiple comparisons that were corrected for multiplicity using the Dunnett's correction for comparisons against the control. In the ANCOVA model, the fixed effect of disease state was coded as an ordered factor which allowed the evaluation of linear trends. As mentioned earlier, between group multiple comparisons were conducted using the Dunnett's test for comparisons against the control group. All analyses were conducted using R [48] under Rstudio. Details of the specific analysis and data presentation are included in the figure legends. All raw data and box plots graphs for this study has been deposited into the following site: dx.doi.org/10.17504/protocols.io.bhwuj7ew.

## Results

### Age and sex distribution of subjects in Cohort 1 and Cohort 2

The demographic characteristics of subjects for Cohorts 1 and 2 are shown in Table 1.

Representative fundus color photographs and optical coherence tomographs (OCT) of patients in this study are shown in Fig 1B–1H. Color fundus images from intermediate dry AMD (AREDS stage 3) show scattered and confluent soft (fluffy yellowish deposits, large arrows) and hard drusen (well-demarcated deposits, small arrows) in the macula (Fig 1B and 1C). Central GA defined as loss of outer retina, RPE and CC (Fig 1D; outlines shown in small arrows with solid black outlines) with their corresponding fundus autofluorescence images (Fig 1E). Another example of GA is shown in Fig 1F and 1G. Areas of hypo-autofluorescence correspond to areas of GA, representing loss of RPE autofluorescence in response to blue light. A cross-section OCT image of the fovea (Fig 1H) from Fig 1F and 1G shows loss of outer retinal, RPE and CC (between arrows with solid black outlines).

### Stepwise approach to identification of biomarkers that are altered in GA in Cohort 1

The methodology for identification of biomarkers is shown in Fig 1A. Ninety-four plasma samples in Cohort 1 were evaluated using the Myriad Luminex xMAP technology. Summary statistics for these biomarkers are shown in S2 Table. Of the 320 biomarkers evaluated, 15% had levels that were below the LLOQ or had QC values that were above acceptable limits. The remaining 273 markers from in the first dataset were further evaluated using Kruskal-Wallis, PCA, and PLS analyses. This rendered a total of 102 markers that showed statistically significant changes. These 102 markers were then submitted to a MANOVA corrected for multiplicity leading to 83% enrichment and identification of 56 markers that remained significantly altered in the dataset (Fig 1A; S2 Table, yellow highlights). Biomarkers that showed significant differences were then analyzed using univariate analysis of covariance (ANCOVA) correcting for age. Of the 56 original biomarkers identified as changing in the original MANOVA, 40 markers (including Aβ(1–40)) reached statistical significance in the univariate ANCOVAs (see

**Table 1. Demographic characteristics of study subjects in Cohort 1 and Cohort 2.**

| Cohort 1 | | Control (N = 33) | AREDS 3 (N = 24) | GA (N = 37) | Total (N = 94) |
|---|---|---|---|---|---|
| Age (years) | | | | | |
| | Mean | 72.4 | 79.4 | 80.5 | 77.0 |
| | SD | 7.71 | 5.56 | 8.79 | 8.29 |
| | Median | 71 | 79 | 84 | 78 |
| | Min.–Max. | 61–89 | 64–87 | 56–97 | 56–97 |
| Sex | Female | 20 (61%) | 20 (83%) | 23 (62%) | 63 (67%) |
| | Male | 13 (39%) | 14 (13%) | 14 (38%) | 31 (33%) |
| Ethnicity | Hispanic/Latino | 1 (3%) | 0 (0%) | 1 (3%) | 2 (2%) |
| | Non-Hispanic White | 32 (97%) | 23 (100%) | 37 (97%) | 92 (98%) |
| **Cohort 2** | | **Control (N = 29)** | **AREDS 3 (N = 22)** | **GA (N = 35)** | **Total (N = 86)** |
| Age (years) | | | | | |
| | Mean | 76.8 | 76.3 | 76.9 | 76.7 |
| | SD | 6.14 | 7.16 | 7.19 | 6.96 |
| | Median | 77 | 76 | 77 | 77 |
| | Min.–Max. | 65–89 | 61–89 | 62–92 | 61–89 |
| Sex | Female | 24 (83%) | 15 (68%) | 17 (49%) | 56 (65%) |
| | Male | 5 (17%) | 7 (32%) | 18 (51%) | 30 (35%) |
| Ethnicity | Hispanic/Latino | 0 (0%) | 0 (0%) | 2 (6%) | 2 (2%) |
| | Non- Hispanic White | 29 (100%) | 22 (100%) | 33 (94%) | 84 (98%) |

There were 94 subjects in Cohort 1, with 63 females and mean age of 77.0 years. Cohort 1 was composed of 33 non-AMD controls, 24 AREDS 3 and 37 GA. In the GA group, 17 were recruited from the practice of KL and 20 were randomly selected from GSK multi-center trial (Clinicaltrials.gov identifier NCT01342926; GSK study 114341). In Cohort 1, there were statistically significant differences in ages in women, but not for men for AREDS 3 and GA vs. the Control group (each, P<0.05; S1 Fig). In Cohort 2, there were 86 subjects with 56 females and mean age of 76.7 years. Cohort 2 was composed of 29 non-AMD controls, 22 AREDS 3 and 35 GA. Among the GA group, 3 were recruited from the practice of KL and 32 were randomly selected from the GSK multi-center trial mentioned above. There were no statistically significant differences in ages among women and men across the diagnostic categories in this cohort.

S3 Table and S2 Fig). One additional marker, sAPP was analyzed separately by ELISA and reached statistical significance. In contrast, Aβ(1–42) did not reach statistical significance, but was included for comparison with the statistically significant Aβ(1–40) and sAPP.

These 42 markers (including 40 from Cohort 1 plus sAPP and Aβ(1–42) were then re-evaluated in the Cohort 2 using different assay technologies as previously indicated. Data generated were subjected to univariate ANCOVAs as previously discussed. Readable data are shown in S4 Table. In Cohort 2, 11 biomarkers reached statistical significance and were therefore confirmed from Cohort 1. Aβ(1–42) was included as the 12th marker in the analysis for comparison with statistically significant Aβ(1–40) and sAPP. A bioinformatics evaluation was also conducted to determine the potential relationships among them the identified markers (see below).

**Bioinformatics network analysis of biomarkers identified in Cohort 1.** The complete list of analytes and summary statistics for their concentrations in study subjects from Cohort 1 are presented in S2 Table. Aβ(1–42) was also included in this table to contrast its P value with that of Aβ(1–40). A bioinformatic network analysis was performed on the corresponding genes and a subnetwork containing 30 of the input genes identified (Fig 2 and S5 Table).

This network includes several strongly connected gene hubs, including MMP9 (mixed metalloproteinase), CD40 (cluster differentiation), IL1B (interleukin-1β), SPP1 (osteopontin) and APP. Twelve of the genes corresponding to differentially expressed biomarkers were directly or indirectly (via transcriptional relationships) connected to APP (Fig 2), the protein that leads to production of Aβ.

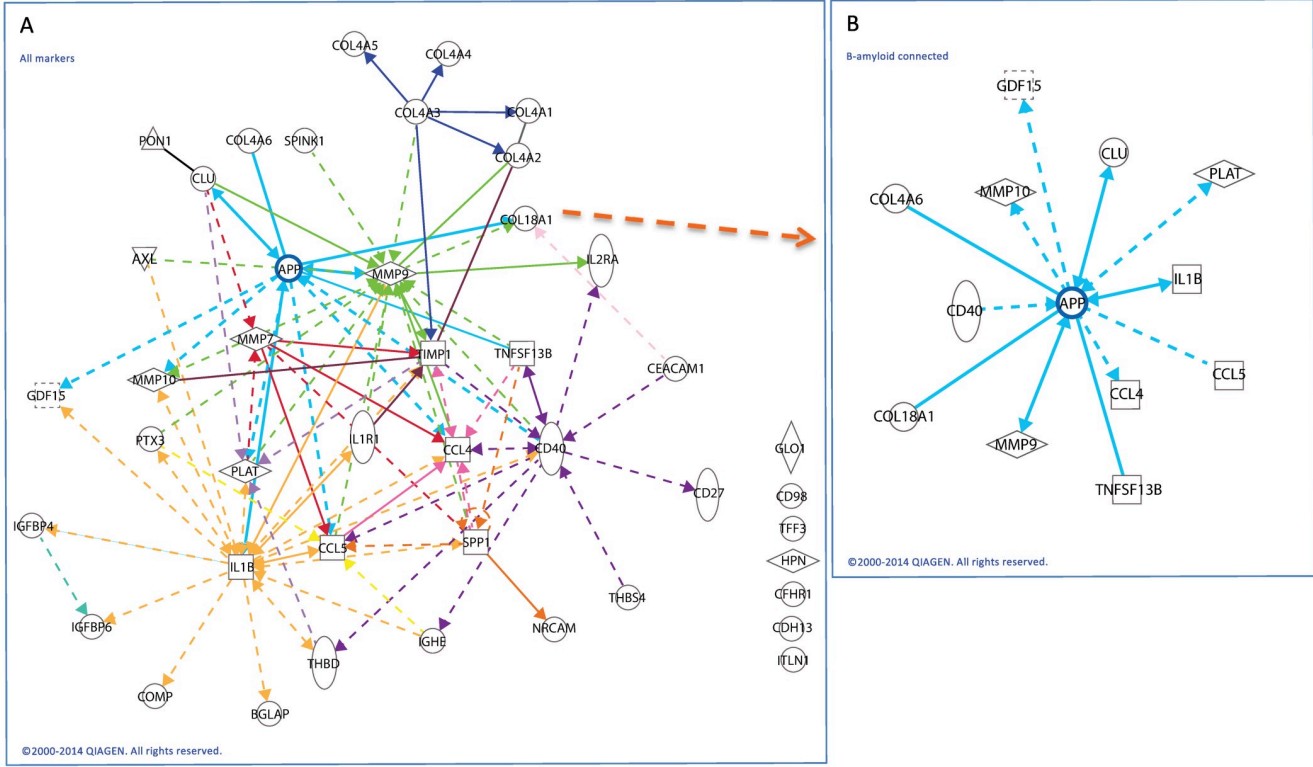

**Fig 2. Network connections between biomarkers detected in Cohort 1 and the amyloid pathway.** (A) Network analysis of genes corresponding to analytes that were statistically significant for GA in Cohort 1 were identified a sub-network containing many of the factors (S3 Table). The network was centered around hub genes, one of which was amyloid precursor protein (APP; blue lines). The nomenclature of significant proteins and genes identified in this network are shown in S5 Table. (B) Network shows genes with direct or indirect (transcriptional) relationship with APP. Some of these biomarkers were identified in Cohort 1 and confirmed in Cohort 2 (refer to text). Solid lines indicate direct relationships; dotted lines indicate indirect (transcriptional) relationships.

## Statistically significant plasma biomarkers identified in Cohort 2

Forty-two markers (including sAPP and Aβ(1–42)) identified in Cohort 1 were then re-analyzed in Cohort 2 using different analytical methodologies. Eleven markers were statistically different in GA vs. Control and AREDS 3 groups. These include omentin, pentraxin-3, metalloproteinase (MMP) -9, insulin-like growth factor binding protein (IGFBP) -6, AXL receptor tyrosine kinase, tissue plasminogen activator (tPA), tumor-associated trypsinogen inhibitor (TATI), MMP-7, sAPP, Aβ(1–40), and osteopontin. Aβ(1–42) was analyzed as the 12[th] marker and was not statistically significant. Summary statistics for all biomarker concentrations for Cohort 2 are shown in S4 Table.

   **Plasma levels of sAPP and amyloid-β(1–40) but not amyloid-β(1–42) are elevated in GA.**   We measured levels of two major amyloid constituents, Aβ(1–40) and Aβ(1–42), as well

as levels of the soluble parent molecule, sAPP (Fig 3). In both cohorts, plasma levels of Aβ(1–40) (Fig 3A and 3B) and sAPP (Fig 3E and 3F), but not Aβ(1–42) (Fig 3C and 3D) were significantly elevated in GA versus (vs.) Control (*; Aβ(1–40)) and GA vs. AREDS 3 and Control

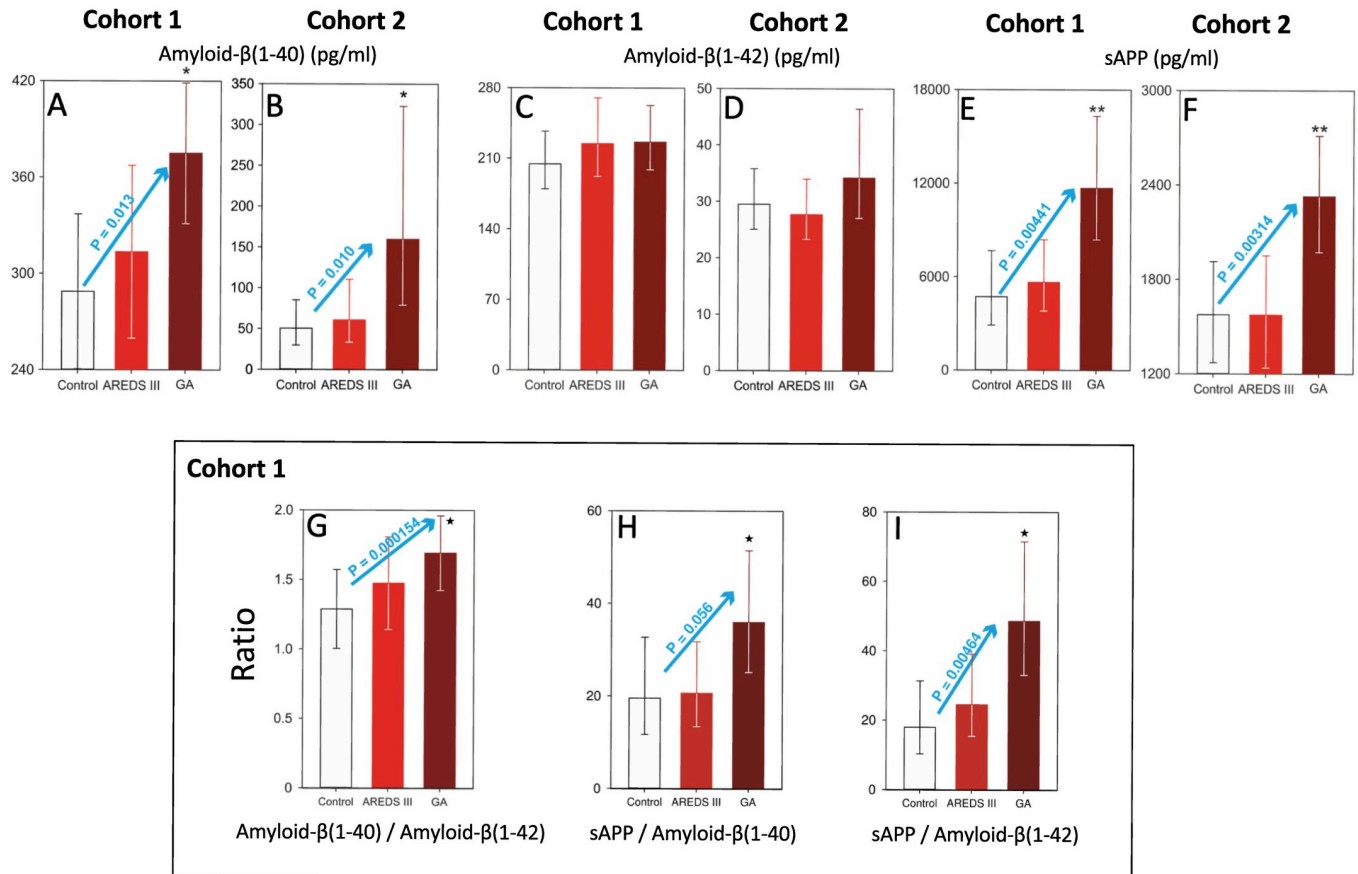

**Fig 3. Plasma levels of Aβ(1–40), Aβ(1–42) and sAPP and their association with geographic atrophy in Cohorts 1 and 2.** (**A**) Data are presented as the least squares means ± 95% confidence limits. Analysis of covariance using age as the covariate followed by the Tukey test on raw data. There is a statistically significant linear trend (P<0.013; blue arrow) among the 3 groups. (**B**) Data are presented as geometric least squares means ± 95% confidence limits. Analysis of covariance followed by the Tukey test on the base 10 logarithm of the data. There is a statistically significant linear trend (P<0.010; blue arrow) among the 3 groups. (**C, D**) Data are presented as the least squares means ± 95% confidence limits. Analysis of covariance using age as the covariate followed by the Tukey test on the inverse of the data. (**E**) Data are presented as the least squares means ± 95% confidence limits. Analysis of covariance using age as the covariate followed by the Tukey test on raw data. There is a statistically significant linear trend (P<0.00441; blue arrow) among the 3 groups. (**F**) Data are presented as geometric least squares means ± 95% confidence limits. Analysis of covariance followed by the Tukey test on the square root of the data. There is a statistically significant linear trend (P<0.00314, blue arrow) among the 3 groups. Asterisk (*), statistically significant differences (P < 0.05) versus the control group. Double asterisk (**), statistically significant differences (P<0.05) versus control and AREDS 3 groups. The Y axes in these graphs have different scales. AREDS III, intermediate dry AMD; GA, geographic atrophy. **Box: Plasma ratios of Aβ(1–40)/Aβ(1–42) and sAPP/Aβ.** (**G**) Data are presented as least squares means ± 95% confidence limits. Analysis of covariance followed by the Tukey test on the raw data. There is a statistically significant linear trend (P<0.000154; blue arrow) among the 3 groups. (**H**) Data are presented as geometric least squares means ± 95% confidence limits. Analysis of covariance on base 10 logarithm of the data followed by the Tukey test on the raw data. There is no statistically significant linear trend (P<0.056, blue arrow) among the 3 groups. (**I**) Data are presented as geometric least squares means ± 95% confidence limits. Analysis of covariance on base 10 logarithm of the data followed by the Tukey test on the raw data. There is a statistically significant linear trend (P<0.00464, blue arrow) among the 3 groups. Asterisk (*) indicate statistically significant differences (P < 0.05) versus the control group. The Y axes in these graphs have different scales. AREDS III, intermediate dry AMD; GA, geographic atrophy.

(**; sAPP) (Fig 3A and 3F; P<0.05 for all GA subgroups). Moreover, in both cohorts, there was a statistically significant linear trend among the 3 diagnostic groups for Aβ(1–40) (Fig 3A and 3B; P = 0.013 and P = 0.010, respectively) and for sAPP (Fig 3E and 3F; blue arrows, P = 0.00441 and P = 0.00314, respectively). Data from Cohort 1 was further analyzed to determine whether ratios of Aβ/Aβ and sAPP/Aβ were also significantly altered in GA (as it has also been reported in Alzheimer's disease) [49]. Fig 3G and 3I show that ratios of Aβ(1–40)/Aβ(1–42) (Fig 3G), sAPP/Aβ(1–40) (Fig 3H) and sAPP/Aβ(1–42) (Fig 3I) are elevated for the GA vs. Control in all subgroups (*, P < 0.05). Furthermore, there was a statistically significant trend among the diagnostic subgroups for Aβ(1–40)/Aβ(1–42) and sAPP/Aβ(1–42) ratios (blue arrows, P = 0.000154 and P = 0.00464, respectively). Overall these observations suggest an intimate association between GA and the amyloid pathway, which is not reflected in intermediate stage dry AMD (AREDS 3) and control samples.

**Identification of plasma biomarkers associated with disturbances of the extracellular matrix and tissue homeostasis.** Our analysis of plasma analytes that are associated with changes in the extracellular matrix (ECM) and loss of homeostasis using the aforementioned methodology, yielded other plasma markers besides those associated with the amyloid pathway. These biomarkers were arbitrarily divided into non-kinase enzymatic factors and others. Biomarkers with enzymatic properties that were significantly elevated for GA vs. Control (*; P < 0.05) or GA vs. Control and AREDS 3 (**; P < 0.05) are shown in Fig 4.

These biomarkers include tPA (Fig 4A and 4B), TATI (Fig 4C and 4D) and MMP-9 (Fig 4G and 4H). Plasma levels of MMP-7 were significantly decreased in both cohorts as compared to those of Control and AREDS 3 groups (**; P < 0.05) (Fig 4E and 4F). There were linear trends across the diagnostic groups for Cohorts 1 and 2 for tPA (blue arrows, P = 0.0001 and P = 0.0001, respectively; Fig 4A and 4B), TATI (blue arrows, P = 0.011 and P = 0.003, respectively; Fig 4C and 4D), MMP-7 (blue arrows, P = 0.003 and P = 0.004, respectively; Fig 4E and 4F), and MMP-9 (blue arrows, P = 0.001 and P = 0.02, respectively; Fig 4G and 4H).

Plasma levels of non-enzymatic factors related to changes in the ECM and homeostasis that were significantly altered in GA vs. Control (*; P < 0.05) or GA vs. Control and AREDS 3 (**; P < 0.05) are shown in Fig 5. These include IGFBP6 (Fig 5A and 5B), AXL (Fig 5C and 5D), omentin (Fig 5E and 5F), pentraxin-3 (Fig 5G and 5H) and osteopontin (Fig 5I and 5J). Statistically significant linear trends were detected across the groups for Cohorts 1 and 2 for all these factors: IGFBP6 (blue arrows, P < 0.0001 and P < 0.0001, respectively; Fig 5A and 5B), AXL (blue arrows, P < 0.005 and P < 0.0001, respectively; Fig 5C and 5D); omentin (blue arrows, P < 0.0001 and P < 0.0001, respectively; Fig 5E and 5F); pentraxin-3 (blue arrows, P < 0.0001 and P < 0.0001, respectively; Fig 5G and 5H) and osteopontin (blue arrows, P < 0.019 and P < 0.004, respectively; Fig 5I and 5J).

## Discussion

As the prevalence of AMD increases with population expansion, 2 forms of advanced AMD equally contribute to vision loss: GA and neovascular AMD. Currently neovascular AMD is treated with anti-VEGF therapy, but to date, all late-stage clinical trials for GA have failed to identify any effective agent. GA exhibits an unusual but predictable pathology, in which there is a transition zone between normal retinal tissue and an adjacent area of progressive tissue loss, manifested by disorganization of the RPE monolayer, increasing basolinear deposits, and shedding of PRs. This transition zone is continuous with the center of the GA lesion, in which complete apoptosis and loss of the PR-RPE-CC complex is observed [9]. The area of RPE loss is easily identified by imaging modalities such as fundus autofluorescence and NIR imaging, which may be used as a clinical endpoint for conducting clinical trials [50]. The etiology of the

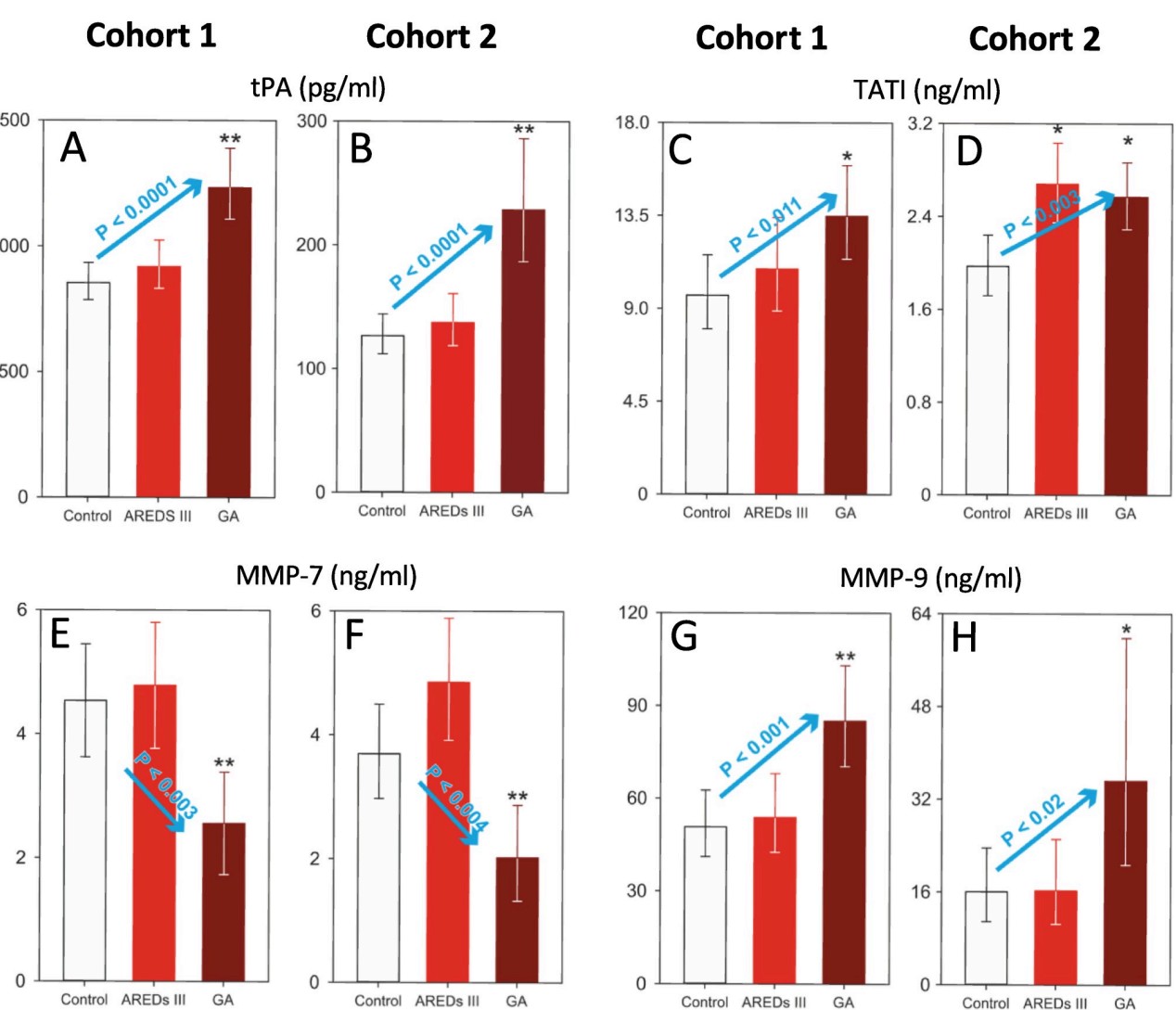

**Fig 4. Plasma levels of enzymatic biomarkers associated with geographic atrophy in Cohorts 1 and 2.** (**A, B**) Data are presented as the geometric least squares means ± 95% confidence limits. Analysis of covariance using age as the covariate followed by the Tukey test on the inverse of the data. There is a statistically significant linear trend in Cohort 1 (**A**; P < 0.0001; blue arrow) and Cohort 2 (**B**; P < 0.0001, blue arrow). (**C**) Data are presented as the geometric least squares means ± 95% confidence limits. Analysis of covariance using age as the covariate followed by the Tukey test on the base 10 logarithm of the data. There is a statistically significant linear trend (P < 0.011, blue arrow) among the 3 groups. (**D**) Data are presented as the geometric least squares means ± 95% confidence limits. Analysis of covariance using age as the covariate followed by the Tukey test on the square root of the data. There is a statistically significant linear trend (P < 0.003, blue arrow) among the 3 groups. (**E**) Data are presented as the geometric least squares means ± 95% confidence limits. Analysis of covariance using age as the covariate followed by the Tukey test on the base 10 logarithm of the data. There is a statistically significant linear trend (P < 0.003, blue arrow) among the 3 groups. (**F**) Data are presented as the geometric least squares means ± 95% confidence limits. Analysis of covariance using age as the covariate followed by the Tukey test on the raw data. There is a statistically significant linear trend (P < 0.004, blue arrow) among the 3 groups. (**G, H**) Data are presented as the geometric least squares means ± 95% confidence limits. Analysis of covariance using age as the covariate followed by the Tukey test on the base 10 logarithm of the data. There is a statistically significant linear trend in both Cohort 1 (**G**; P < 0.001, blue arrow) and Cohort 2 (**H**; P < 0.02, blue arrow) among the 3 groups. Asterisk (*), statistically significant differences (P < 0.05) versus the control group. Double asterisk (**), statistically significant differences (P<0.05) versus control and AREDS 3 groups. The Y axes in these graphs have different scales. AREDS III, intermediate dry AMD; GA, geographic atrophy.

transition zone in GA is not known and numerous drugs targeting a variety of pathways whose activities are suspected to play a role in pathogenesis of GA have failed in late stage clinical trials [21].

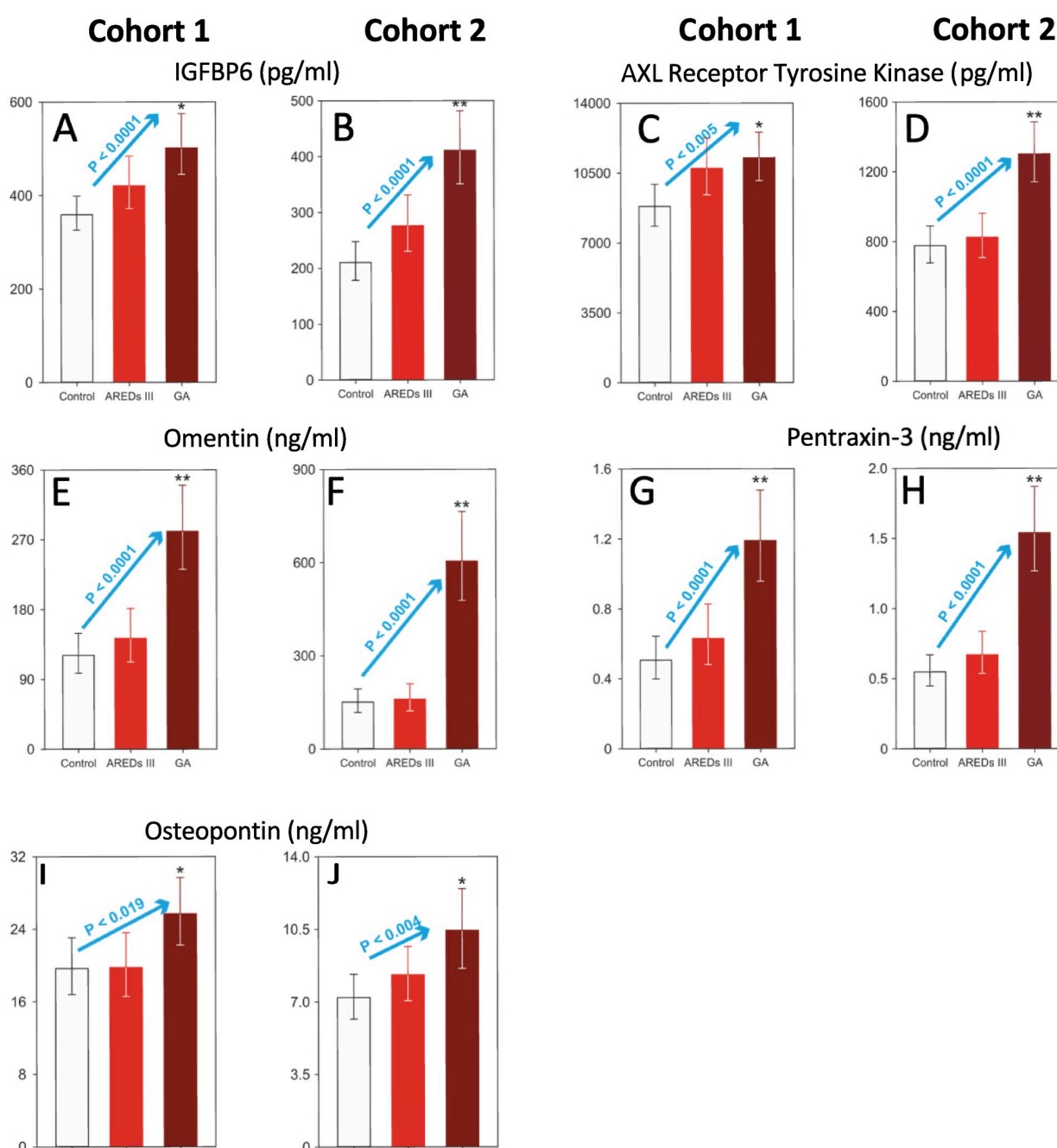

**Fig 5. Plasma levels of non-enzymatic biomarkers associated with geographic atrophy in Cohort 1 and Cohort 2.** (**A**) Data are presented as the geometric least squares means ± 95% confidence limits. Analysis of covariance using age as the covariate followed by the Tukey test on the inverse of the data. There is a statistically significant linear trend (P < 0.0001, blue arrow) among the 3 groups. (**B**) Data are presented as the geometric means ± 95% confidence limits. Analysis of covariance using age as the covariate followed by the Tukey test on the base 10 logarithm of the data. There is a statistically significant linear trend (P < 0.0001, blue arrow) among the 3 groups. (**C, D**) Data are presented as the geometric least squares means ± 95% confidence limits. Analysis of covariance using age as the covariate followed by the Tukey test on the base 10 logarithm of the data. There is a statistically significant linear trend in Cohort 1 (**C**; P < 0.005, blue arrow) and Cohort 2 (**D**; P < 0.001, blue arrow) among the 3 groups. (**E, F**) Data are presented as the geometric least squares means ± 95% confidence limits. Analysis of covariance using age as the covariate followed by the Tukey test on the base 10 logarithm of the data. There is a statistically significant linear trend in Cohort 1 (**E**; P < 0.0001, blue arrow) and Cohort 2 (**F**; P < 0.001, blue arrow) among the 3 groups. (**G, H**) Data are presented as the geometric least squares means ± 95% confidence limits. Analysis of covariance using age as the covariate followed by the Tukey test on the base 10 logarithm of the data. There is a statistically significant linear trend in Cohort 1 (**G**; P < 0.001, blue arrow) and Cohort 2 (**H**; P < 0.001, blue arrow) among the 3 groups. (**I, J**) Data are presented as the geometric least square means ± 95% confidence limits. Analysis of covariance using age as the covariate followed by the Tukey test on the base 10

logarithm of the data. There is a statistically significant linear trend in Cohort 1 (**I**; P < 0.019, blue arrow) and Cohort 2 (**J**; P < 0.004, blue arrow) among the 3 groups. Asterisk (*), statistically significant differences (P < 0.05) versus the control group. Double asterisk (**), statistically significant differences (P<0.05) versus control and AREDS III groups. The Y axes in these graphs have different scales. AREDS III, intermediate dry AMD; GA, geographic atrophy.

Understanding the pathways that may be critically involved in development and progression of GA would be key for designing novel therapeutic approaches to the disease. Identification of systemic biomarkers associated with GA may provide clues as to which pathways are involved in the pathogenic process. This study was designed to identify plasma biomarkers that were associated specifically with GA as compared to intermediate dry AMD and non-AMD controls. Using a wide spectrum Luminex-based approach developed by Myriad RBM, we evaluated 320 (+ sAPP by ELISA) analytes in the first cohort (Cohort 1) and identified 40 (+ sAPP) biomarkers that were significantly altered in GA. We then analyzed these biomarkers with bioinformatics and showed that most of them (shown in Fig 2) were integrated and centered over the amyloid pathway. Interestingly, in Cohort 1, sAPP and Aβ(1–40) levels were significantly elevated in GA, while Aβ(1–42) was surprisingly not significantly altered. We subsequently re-evaluated these 42 factors (including sAPP and Aβ(1–42)) using different methodologies in Cohort 2. Among these biomarkers, we identified 11 as significantly altered in GA. Again, our data shows that sAPP and Aβ(1–40) but Aβ(1–42) were elevated in Cohort 2. Our studies clearly show that a set of specific systemic biomarkers are associated with GA and that GA may have a plasma signature characterized by 11 biomarkers as shown in Figs 3–5.

It has been shown that Aβ is a constituent of drusen [25] and contributes to induction and maintenance of a local inflammatory state [27, 28]. The upstream molecule, APP, undergoes extensive processing, and mutations in APP are associated with familial AD [34]. Cleavage of APP by α-secretase results in production of sAPPα and p3 (by γ-secretase), where sAPPα is involved in the non-amyloidogenic pathway. This pathway supports the health and proliferation of resident stem cells [35]. Alternatively, cleavage of APP by β-secretase results in the amyloidogenic sAPPβ and downstream production of Aβs by γ-secretase. Aβ moieties are highly neurotoxic and abundant in senile plaques in AD [36]. Other than its presence in drusen deposits associated with AMD, the role of the Aβ pathway has not been thoroughly investigated in GA. In both of our study cohorts, both sAPP and Aβ(1–40) but not Aβ(1–42) were significantly elevated in GA as compared to AREDS 3 and non-AMD controls (Fig 3A–3F). Interestingly, both Aβ(1–40) and Aβ(1–42) have been associated with AD [36]. Although Aβ(1–40) is more abundant, Aβ(1–42) readily forms oligomers. Aβ(1–42)-derived oligomers, dimers, and fibrillar material that have been isolated from the AD brain and CSF, appear more closely with, and anatomically correlate with the disease process [36]. The ratios of Aβ(1–40)/Aβ(1–42), sAPP/Aβ(1–40) and sAPP/Aβ(1–42) were also significantly elevated in plasma of subjects with GA vs. AREDS 3 and non-AMD control (Fig 3G–3I). Interestingly, the ratio of Aβ(1–42)/Aβ(1–40) has been studied as a plasma biomarker in AD [49]. Fandos et al. (2017) [49] showed an inverse association between plasma Aβ(1–42)/Aβ(1–40) ratio and cortical Aβ deposition and Aβ burden. In parallel to this observation, we assessed whether there were any associations between drusen and RPD burden, size and shape of the GA lesions, and presence or absence of peripheral halos by OCT and FAF using the Heidelberg Spectralis system. Our analysis did not show any associations between Aβ or Aβ ratio and the aforementioned imaging markers. It is not clear to us why plasma levels of Aβ(1–40) and not Aβ(1–42) are associated with GA, which is again in direct contrast with their associations in AD. In fact, most studies have implicated Aβ(1–42) as the key damaging factor in brain lesions of AD [49].

Aβ(1–40) may be a key biomarker to distinguish between the pathological processes associated with GA from those that occur in AD. We have recently shown that Aβ (both 40 and 42) can inhibit the activity of CFI, one of the key breaks in the alternative complement pathway, leading to a shift in overproduction of C3b in lieu of inactive pathway via iC3b. Increased C3b leads to eventual production of C5b678(9) membrane attack complex [30]. This relationship between Aβ and CFI may tie in the putative roles of upregulated alternative complement pathway in the etiology of GA [30]. Aβ(1–40) appears more potent at inhibiting CFI bioactivity than Aβ(1–42) [30]. This difference in the biological activity of Aβ(1–40) to inhibit CFI bioactivity may be implicated in the mechanisms leading to an exacerbated activation of the alternative complement cascade locally at the level of the RPE/Bruch's membrane interphase. (See [30] for a more in-depth discussion on this topic). In the amyloidogenic pathway, sAPPβ cleaved from APP by β-secretase is then cleaved by γ-secretase to render a 48 or 49 fragment (Aβ(1–48) and Aβ(1–49)). These fragments are then subjected to carboxylase digestion by γ-secretase. Aβ(1–48) is rendered into Aβ(1–38) and Aβ(1–42), while Aβ(1–49) is rendered into Aβ(1–40) and Aβ(1–43) [51]. Based on our observations, we postulate that the carboxylase activity of γ-secretase in GA may preferentially direct splicing of sAPPβ to Aβ(1–40) over Aβ(1–42). These findings are in agreement with those of Inoue et al. (2017) [51], who using CRISPR/Cas9-mediated gene activation of APP or β-secretase in fibroblasts derived from familial AD, unmasked a processing defect in γ-secretase, explaining the preferential processing of carboxylase activity and increased levels of Aβ(1–42) [51].

We also observed that sAPP was also elevated and strongly associated with GA. It is important to note that our ELISA antibody does not distinguish between sAPP and sAPPβ (but distinguishes between sAPP and sAPPα). Detection of elevated levels of sAPP may therefore indirectly suggest elevated levels of amyloidogenic sAPPβ. In parallel to this observation, Cuchillo-Ibanez et al. (2015) [32] have detected full length sAPP in cerebrospinal fluid of AD and implicated it in the disease process.

We arbitrarily grouped the non-amyloidogenic biomarkers into enzymatic and non-enzymatic groups (Figs 4 and 5). These biomarkers overall are indirect indicators of remodeling of the ECM and a general loss of homeostasis within the PR-RPE-CC complex. Factors such as TATI, MMPs, tPA, omentin and pentraxins activate various enzymatic processes help degrade the Bruch's membrane and RPE monolayer, and promote disruption of the normal anatomical barriers within the PR-RPE-CC complex, all hallmarks of GA [52–57]. Activities of MMPs (MMP-1, 2 and 9) have been specifically implicated in advanced AMD, both in angiogenesis and in disruption of the RPE monolayer [58, 59]. Interestingly, plasma levels of MMP-7 were reduced in GA, while levels of MMP-9 were increased in both of our study cohorts (Fig 4). The imbalance between TIMPs and MMPs have been implicated in advanced AMD including GA [60]. Other non-enzymatic factors including IGFBP6, AXL and osteopontin are associated with inflammatory, migratory and proliferative responses to injury as seen in GA [61–63]. Interestingly, osteopontin is a component of drusen and basal laminar deposits [64].

## Conclusions

Plasma levels collected from 2 separate cohorts of subjects with GA, intermediate dry AMD and non-AMD controls were evaluated using a large multiplex library and subsequently in a second cohort by individual ELISA or multiplexing. Findings confirm that subjects with GA express markers associated with the β-amyloid pathway and other factors related to loss of ECM and tissue homeostasis. Elevations of the sAPP and Aβ(1–40), but not Aβ(1–42) were detected in both cohorts, suggesting that there is a preferential processing of APP downstream peptides to the amyloidogenic pathway. Increased levels of Aβ(1–40) over Aβ(1–42) suggest a

potential preferential role for the carboxylase activity of the γ-secretase protein that leads to production of Aβ(1–40). These observations are in direct contrast with those made in AD, in which Aβ(1–42) is the more prevalent amyloid. A series of other proteins and enzymes were identified to be associated with GA, and also associated with sAPP using bioinformatics and protein association studies. These factors are overall indications of remodeling of the ECM and loss of homeostasis of the PR-RPE-CC complex, as it is implicated in AMD. Association of the β-amyloid pathway with GA may suggest novel disease pathways beyond the known generalized deleterious effects of β-amyloid. Altered activity of the carboxylase unit of γ-secretase protein may possibly provide novel treatment targets for GA.

## Supporting information

**S1 Fig. Distribution of study subjects in Cohort 1 and Cohort 2.** Data are presented as the least squares means ± 95% confidence limits. Analysis was conducted by a two-way analysis of variance followed by the Tukey test on the raw data. Asterisks (*) indicate statistically significant differences (P < 0.05) versus women control group in Figure A. (Cohort 1). The Y axes are different in each graph.
(TIF)

**S2 Fig. Plasma levels of 42 biomarkers associated with geographic atrophy in Cohort 1.** Asterisk (*), statistically significant differences (P < 0.05) versus the control group. Double asterisk (**), statistically significant differences (P<0.05) versus control and AREDS III groups. The Y axes in these graphs have different scales. Please refer to Fig 3 in the manuscript for Aβ (1–40), Aβ(1–42) and sAPP. Key: Control, non-AMD control; AREDS 3, intermediate dry AMD; GA, geographic atrophy. (**A**) Pon1. Data represent the fitted means ± SE. Analysis of covariance using age as the covariate followed by the Tukey test on the raw data. (**B**) Omentin. Data represent the fitted geometric means ± SE estimated by the Taylor series expansion. Analysis of covariance using age as the covariate followed by the Tukey test on the base 10 logarithm of the data. (**C**) ST2. Data represent the fitted geometric means ± SE estimated by the Taylor series expansion. Analysis of covariance using age as the covariate followed by the Tukey test on the base 10 logarithm of the data. (**D**) T-Cadherin. Data represent the fitted means ± SE estimated by the Taylor series expansion. Analysis of covariance using age as the covariate followed by the Tukey test on the inverse of the data. (**E**) Pentraxin-3. Data represent the fitted geometric means ± SE estimated by the Taylor series expansion. Analysis of covariance using age as the covariate followed by the Tukey test on the base 10 logarithm of the data. (**F**) tPA. Data represent the fitted means ± SE estimated by the Taylor series expansion. Analysis of covariance using age as the covariate followed by the Tukey test on the inverse of the data. (**G**) LGL. Data represent the fitted geometric means ± SE estimated by the Taylor series expansion. Analysis of covariance using age as the covariate followed by the Tukey test on the base 10 logarithm of the data. (**H**) Baff. Data represent the fitted geometric means ± SE estimated by the Taylor series expansion. Analysis of covariance using age as the covariate followed by the Tukey test on the base 10 logarithm of the data. (**I**) IgE. Data represent the fitted means ± SE. Analysis of covariance using age as the covariate followed by the Tukey test on inverse of the data. (**J**) MIP-1β. Data represent the fitted geometric means ± SE estimated by the Taylor series expansion. Analysis of covariance using age as the covariate followed by the Tukey test on the base 10 logarithm of the data. (**K**) NrCAM. Data represent the fitted geometric means ± SE estimated by the Taylor series expansion. Analysis of covariance using age as the covariate followed by the Tukey test on the base 10 logarithm of the data. (**L**) CFHR1. Data represent the fitted geometric means ± SE. Analysis of covariance using age as the covariate followed by the Tukey test on the raw data. (**M**) MMP-9. Data represent the fitted geometric

means ± SE estimated by the Taylor series expansion. Analysis of covariance using age as the covariate followed by the Tukey test on the base 10 logarithm of the data. (**N**) C1QR1. Data represent the fitted geometric means ± SE estimated by the Taylor series expansion. Analysis of covariance using age as the covariate followed by the Tukey test on the base 10 logarithm of the data. (**O**) CLU. Data represent the fitted geometric means ± SE estimated by the Taylor series expansion. Analysis of covariance using age as the covariate followed by the Tukey test on the base 10 logarithm of the data. (**P**) IGFBP6. Data represent the fitted means ± SE estimated by the Taylor series expansion. Analysis of covariance using age as the covariate followed by the Tukey test on the inverse of the data. (**Q**) MMP-7. Data represent the fitted geometric means ± SE estimated by the Taylor series expansion. Analysis of covariance using age as the covariate followed by the Tukey test on the base 10 logarithm of the data. (**R**) GDF-15. Data represent the fitted geometric means ± SE estimated by the Taylor series expansion. Analysis of covariance using age as the covariate followed by the Tukey test on the base 10 logarithm of the data. (**S**) CD27. Data represent the fitted geometric means ± SE estimated by the Taylor series expansion. Analysis of covariance using age as the covariate followed by the Tukey test on the base 10 logarithm of the data. (**T**) Osteocalcin. Data represent the fitted geometric means ± SE estimated by the Taylor series expansion. Analysis of covariance using age as the covariate followed by the Tukey test on the raw data. (**U**) RANTES. Data represent the fitted geometric means ± SE estimated by the Taylor series expansion. Analysis of covariance using age as the covariate followed by the Tukey test on the base 10 logarithm of the data. (**V**) MMP-10. Data represent the fitted geometric means ± SE estimated by the Taylor series expansion. Analysis of covariance using age as the covariate followed by the Tukey test on the base 10 logarithm of the data. (**W**) Endostatin. Data represent the fitted geometric means ± SE estimated by the Taylor series expansion. Analysis of covariance using age as the covariate followed by the Tukey test on the base 10 logarithm of the data. (**X**) Ceacam1. Data represent the fitted means ± SE estimated by the Taylor series expansion. Analysis of covariance using age as the covariate followed by the Tukey test on the inverse of the data. (**Y**) Hepsin. Data represent the fitted means ± SE estimated by the Taylor series expansion. Analysis of covariance using age as the covariate followed by the Tukey test on the inverse of the data. (**Z**) TTF3. Data represent the fitted geometric means ± SE estimated by the Taylor series expansion. Analysis of covariance using age as the covariate followed by the Tukey test on the base 10 logarithm of the data. (**AA**) Collagen-4. Data represent the fitted means ± SE estimated by the Taylor series expansion. Analysis of covariance using age as the covariate followed by the Tukey test on the inverse of the data. (**AB**) IL-1r1. Data represent the fitted means ± SE. Analysis of covariance using age as the covariate followed by the Tukey test on the raw data. (**AC**) AXL. Data represent the fitted geometric means ± SE estimated by the Taylor series expansion. Analysis of covariance using age as the covariate followed by the Tukey test on the base 10 logarithm of the data. (**AD**) IL-1β. Data represent the fitted means ± SE estimated by the Taylor series expansion. Analysis of covariance using age as the covariate followed by the Tukey test on the inverse of the data. (**AE**) IL-2ra. Data represent the fitted geometric means ± SE estimated by the Taylor series expansion. Analysis of covariance using age as the covariate followed by the Tukey test on the base 10 logarithm of the data. (**AF**) Osteopontin. Data represent the fitted geometric means ± SE estimated by the Taylor series expansion. Analysis of variance using age as the covariate followed by the Tukey test on the base 10 logarithm of the data. (**AG**) COMP. Data represent the fitted geometric means ± SE estimated by the Taylor series expansion. Analysis of variance using age as the covariate followed by the Tukey test on the base 10 logarithm of the data. (**AH**) TIMP1. Data represent the fitted geometric means ± SE estimated by the Taylor series expansion. Analysis of variance using age as the covariate followed by the Tukey test on the inverse of the data. (**AI**) TATI. Data represent the fitted geometric means ± SE

estimated by the Taylor series expansion. Analysis of variance using age as the covariate followed by the Tukey test on the base 10 logarithm of the data. (**AJ**) TM. Data represent the fitted geometric means ± SE estimated by the Taylor series expansion. Analysis of variance using age as the covariate followed by the Tukey test on the base 10 logarithm of the data. (**AK**) CD40. Data represent the fitted geometric means ± SE estimated by the Taylor series expansion. Analysis of variance using age as the covariate followed by the Tukey test on the inverse of the data. (**AL**) IGFBP4. Data represent the fitted geometric means ± SE estimated by the Taylor series expansion. Analysis of variance using age as the covariate followed by the Tukey test on the inverse of the data. (**AM**) Thrombospondin-4. Data represent the fitted geometric means ± SE estimated by the Taylor series expansion. Analysis of variance using age as the covariate followed by the Tukey test on the base 10 logarithm of the data.
(TIF)

**S1 Table. Cohort 2: Sources and glossary of ELISA and multiplex assays used for measurement of plasma samples.**
(DOC)

**S2 Table. Cohort 1: Summary Statistics (N, mean, standard error of the mean [SE], 25 and 75% percentiles [25% and 75%] and interquartile range [IQR]) of analyte concentrations by group.** YELLOW. Original 56 statistically significant analytes (refer to Fig 1H in the manuscript). BLUE. Aβ(1–42) was not statistically significant. GREEN. sAPP was statistically significant. Key: Control, non-AMD control; AREDS 3, intermediate dry AMD; GA, geographic atrophy.
(DOCX)

**S3 Table. Cohort 1: 40 statistically significant biomarkers for Geographic.** Atrophy and their glossary in order of their P values. The 40 markers were detected by Myriad Luminex xMAP Technology. * sAPP was detected by ELISA and analyzed separately; Aβ(1–42) was not statistically significant but was included for comparison with Aβ(1–40).
(DOCX)

**S4 Table. Cohort 2: Summary statistics (N, mean, standard error of the mean [SE], median [MEDIAN], 25 and 75% percentiles [25% and 75%] and interquartile range [IQR]) of analyte concentrations by group.** Key: Control, non-AMD control; AREDS 3, intermediate dry AMD; GA, geographic atrophy.
(DOCX)

**S5 Table. Cohort 1: Nomenclature of significant proteins and genes identified using bioinformatics.**
(DOCX)

**S1 File.**
(ZIP)

## Acknowledgments

The authors would like to thank Dr. Steve Novick, Department of Biostatistics, GlaxoSmithKline for his support in the statistical analysis.

## Author Contributions

**Conceptualization:** Kameran Lashkari, Gianna C. Teague, Sanjay Kumar, Megan M. McLaughlin, Francisco J. López.

**Data curation:** Kameran Lashkari, Gianna C. Teague, Joanna Betts, Francisco J. López.

**Formal analysis:** Joanna Betts, Francisco J. López.

**Funding acquisition:** Kameran Lashkari, Megan M. McLaughlin, Francisco J. López.

**Investigation:** Kameran Lashkari, Gianna C. Teague, Ursula Beattie, Joanna Betts.

**Methodology:** Kameran Lashkari, Ursula Beattie, Joanna Betts, Megan M. McLaughlin, Francisco J. López.

**Project administration:** Kameran Lashkari, Sanjay Kumar, Megan M. McLaughlin, Francisco J. López.

**Resources:** Joanna Betts.

**Software:** Joanna Betts, Francisco J. López.

**Supervision:** Kameran Lashkari, Francisco J. López.

**Validation:** Kameran Lashkari, Gianna C. Teague, Ursula Beattie, Joanna Betts, Francisco J. López.

**Visualization:** Joanna Betts.

**Writing – original draft:** Kameran Lashkari, Gianna C. Teague, Francisco J. López.

**Writing – review & editing:** Kameran Lashkari, Joanna Betts, Sanjay Kumar, Megan M. McLaughlin, Francisco J. López.

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
