## [Decision Letter · Decision Letter 0]

21 Apr 2020

PONE-D-20-05117

Plasma Biomarkers of the Amyloid Pathway Are Associated with Geographic Atrophy Secondary to Age-Related Macular Degeneration

PLOS ONE

Dear Dr. Lashkari,

Thank you for submitting your manuscript to PLOS ONE. After careful consideration, we feel that it has merit but does not fully meet PLOS ONE’s publication criteria as it currently stands. Therefore, we invite you to submit a revised version of the manuscript that addresses the points raised during the review process.

Both reviewers have highlighted the merit of your work and the scientific questions being investigated, but had some concerns over methodology and interpretation. Please look through their attached comments and assess how best to alleviate their concerns, and I would be very happy to see a revised manuscript.

We would appreciate receiving your revised manuscript by Jun 05 2020 11:59PM. To enhance the reproducibility of your results, we recommend that if applicable you deposit your laboratory protocols in protocols.io, where a protocol can be assigned its own identifier (DOI) such that it can be cited independently in the future. For instructions see: http://journals.plos.org/plosone/s/submission-guidelines#loc-laboratory-protocols

We look forward to receiving your revised manuscript.

Kind regards,

Simon J Clark, D.Phil.

Academic Editor

PLOS ONE

2. Thank you for stating the following in the Funding Statement section:

"This work was supported by Grant #4100112, GlaxoSmithKline, King of Prussia, PA, USA awarded to KL (https://us.gsk.com). JB, SK, MMcM and FJL received funding from GlaxoSmithKline in the form of salaries. The funders of this study had no role in study design, data collection and analysis, decision to publish, or preparation of the manuscript."

4. Please include a copy of Table 4 which you refer to in your text on page 19.

Reviewers' comments:

Reviewer's Responses to Questions

**Comments to the Author**

1. Is the manuscript technically sound, and do the data support the conclusions?

Reviewer #1: Partly

Reviewer #2: Yes

2. Has the statistical analysis been performed appropriately and rigorously? 

Reviewer #1: No

Reviewer #2: N/A

3. Have the authors made all data underlying the findings in their manuscript fully available?

Reviewer #1: No

Reviewer #2: Yes

4. Is the manuscript presented in an intelligible fashion and written in standard English?

Reviewer #1: Yes

Reviewer #2: Yes

5. Review Comments to the Author

Reviewer #1: This a relatively small scale study searching for plasma biomarkers which may indicate the presence of GA as a consequence of AMD.

The rationale for the study is somewhat poorly explained. There is no real need for a predictive biomarker of late-stage disease, since it is readily detectable both from symptomatology and retinal imaging. I take the point that such studies could inform on the underlying biology of disease development, but this manuscript makes no apparent effort to confirm e.g. that the identified circulating biomarkers are indeed also altered in their expression at the back of the eye, and so, while interesting, the study as it stands lacks conviction as to either the mechanistic value of the observed findings, or indeed the value of these proteins as biomarkers of disease progression, rather than just presence.

That said, there is some interesting information in here regarding potential biomarkers. However, I have concerns about the analysis of the two chosen cohorts, especially with respect of the ‘validation’ data presented towards the end of the manuscript, which would need to be addressed before this manuscript could be suitable for publication in PLoS One.

Major comments.

The demographic details for the patient cohorts should be presented in a table. The current presentation, of just age in Figure 4, is both limited and hard to discern detail. Do the authors have further clinical data? Given that their major finding is an increase in proteins commonly linked to Alzheimer’s disease, and that AMD and Alzheimer’s disease share many risk factors (including age, smoking status, BMI), it is important to present these fully, where possible, to rule out that any biomarkers are associated with these factors, rather than AMD per se. The table of ages provided via protocols.io doesn’t state whether each case was in cohort 1 or cohort 2, and has cases in groups labelled wetactive and wetinactive, which are not mentioned in the manuscript?

One would have thought a multiple-testing correction (Benjamini-Hochberg) should be applied to the data from cohort 1? If this is inappropriate, the authors should state why.

Text on line 296 onwards is confusing. ‘Of 56 changing, 42 reached statistical significance’, in other words 14 of the 56 changing weren’t actually changing? Definitions key here, I think. How the authors arrived at this list of 56 is hugely unclear, aside from a list of multiple statistical tests. How were these applied? A table spreadsheet with summary data (not raw data, as requested in the guidance – “For example, in addition to summary statistics, the data points behind means, medians and variance measures should be available.”) from all 273 measured proteins, is available in supplementary table 3, but it’s unclear how proteins were filtered and there is no statistical test result.

The authors then add 2 more molecules, sAPP and the statistically insignificant Ab(1-42), bringing the number of statistically significant biomarkers to 43? I’m unclear how this works.

The bioinformatics network analysis is useful, although somewhat cherry-picked. For example, there seem to be more proteins directly linked to CD40 than to AP. The Ingenuity software produces significance scores as to how over-represented these pathways are in the network – these scores should be reported to support the authors’ conclusions from this analysis.

These 43 were re-evaluated by a different ELISA in a second cohort (although the heading on line 334 states 42), but the manuscript (line 303) then only discusses 12 of them, 11 of which were significant. Does this mean that 30/42 (71%) failed to replicate? Again, far more detail is required, as above, to allow myself, and a reader, to evaluate these data fully.

Box plots, such as described here (https://blog.bioturing.com/2018/05/22/how-to-compare-box-plots/), should be used to illustrate these data, including individual data points as shown for each patient to allow the reader to interpret these data more fully. Units should also be adjusted such that they’re easier to read (e.g. in Supp table 4, Pon1 could be 8.77ug/mL, instead of 8,771,921 pg/mL).

The authors go on to study amyloid beta peptide levels in samples in both cohorts, claiming an increase in the 1-40 form (but not the 1-42 form) in GA. These data are presented in Figure 6. The authors do not address the apparently huge disparity in levels in the control samples between the 2 cohorts (for 1-40 they are ~280pg/ml in cohort 1, and ~50pg/mL in cohort 2). This is critical and impact on the reliability of the data. In my view, y-axes should equalised to make this disparity more apparent.

A similar observation can be made for the non-amyloid biomarkers presented in Figure 7 and 8. What is so different between the cohorts such that, e.g. AXL RTK is 10x lower in controls from cohort 2 than in cohort 1, while other proteins e.g. pentraxin-3 and omentin are (broadly) similar? This is a critical question for this study and the lack of reproducibility between the 2 cohorts, both in terms of identified biomarkers and raw concentrations measured, currently places major doubt on the validity of these findings.

Minor comments:

The abstract should state the methodology used for the initial cohort 1 study i.e. Luminex.

There are numerous typographical errors which need to be corrected.

There are probably too many figures. Figure 1 is redundant – a simple reference will do as this figure is widely used elsewhere. Figures 2 and 3 could be merged into a single figure. Figure 4 should be a table (see above) etc..

Line 305 – the location of where the informatics evaluation took place is irrelevant.

The y axes on figure 4 should be the same, to allow comparison.

Are the p-values on the arrows on Figs 6-8 the significance of the trend, as suggested in the legend? If so, what test was used?

Reviewer #2: Lashkari et al. used a two-stage approach to identify proteins in plasma that were significantly differentially present in patients with early or late AMD compared to healthy controls They identify a set of 11 proteins which are either elevated or reduced in patients. Their approach identified that sAPP and Aß(1-40), but not Aß(1-42) to be elevated in patients with GA, pointing towards and involvement of amyloid pathways in GA pathogenesis. Additional proteins were identified to be dysregulated, particularly those involved in ECM remodelling/homeostasis. I have some comments:

Introduction:

Line 104: These include variants of complement factor (CF) H, CFI, CFB and C3 [13-17]. There are also recent report on a role of C9 and C4A/B in AMD risk and those studies should be cited here.

Methods:

Line 168: Subjects were staged as non-AMD control (AREDS stage 0), intermediate stage dry AMD - AREDS stage 3, and advanced stage dry AMD - GA (AREDS 4/5 [31]).

The cited source has two grading systems, one 9-step and one 6-step grading. While I am no expert on grading, if I remember correctly late stage GA was defined as AREDS grade 9 and 10 (peripheral and central GA, respectively). Did the authors use earlier stages in their assessment which are lacking geographic atrophy? Their fundus images suggest otherwise, with pronounced central GA in Figure 2D and F?

Line 221: Evaluation of biomarkers by ELISA and Multiplexing in Cohort 2

The full list of analytes which were analysed in cohort 2 does not need to be written as such but should rather be in a Table in the supplements and then just cited here.

Line 257: CFI Bioactivity Assay Evaluating Oligomerized A�(1-40) and A�(1-42)

A little bit more detail on the methodology would be good here so the reader does not have to read another manuscript to understand the approach.

Results:

I would like to have a little bite more information on the patient characteristics. A Table summarizing age and other AMD risk factors for all cohorts, stratified by disease status would be good in the beginning.

Table 1:

a) How were the P-values corrected for multiple testing? The table legend mentions multiplicity but no explanation on how that was done.

b) How are the effect sizes (i.e. higher or lower levels in patients compared to controls), standard errors or 95% confidence intervals? The table can be quite large for 56 markers but this should be shown. The legend can be shorter because the proteins are mentioned in Table 2, which can also be moved to the supplement.

Table 3: see comments for Table 2.

Line 289ff: Why did the authors not perform logistic or multinomial regression for their analyses adjusted for age and/or batches/plates for their initial screen in 273 markers? The current approach has two steps and thus more statistical tests, which should be appropriately accounted for.

Line 301ff: The authors need to be clearer about their approach and their statistics. 43 Markers were investigated in cohort 2, but the authors then only talk about 12 markers of which 11 were stat. significant?

Line 308-319: this information can be put into a table, as suggested above and placed before the previous paragraph, since the description of the patient cohort should come first. Otherwise the results are cut in half by this paragraph and the read flow is disrupted.

Line 334ff: See my comments above for lines 301ff

Figure 6D: Why was the inverse of the data used in this analysis?

Discussion:

Line 418ff: This paragraph should rather be in the introduction and not necessarily in the discussion. It does not fit here.

Line 426 to 434: this should also be in the introduction and only briefly summarized here I it is relevant for the results in this study

Line 468: I don’t know if CRISPR/Cas9 was available in 1977.

Line 478: While I appreciate the in-depth discussion of all the markers not involved in amyloid pathways that were identified as significant, this section should be shortened and can be expressed more generally. Clearly the results suggest that remodelling and homeostasis of the ECM seem to be a major theme as well as immune activation and the proteins should be put into this context.

6. PLOS authors have the option to publish the peer review history of their article (what does this mean?). If published, this will include your full peer review and any attached files.

Reviewer #1: No

Reviewer #2: No

---

## [Author Response · Author response to Decision Letter 0]

26 Jun 2020

Please refer to the attached file: Response to Reviewers. Per Reviewers' request we have included a zip file of all Boxplots.

---

## [Decision Letter · Decision Letter 1]

6 Jul 2020

Plasma Biomarkers of the Amyloid Pathway Are Associated with Geographic Atrophy Secondary to Age-Related Macular Degeneration

PONE-D-20-05117R1

Dear Dr. Lashkari,

We’re pleased to inform you that your manuscript has been judged scientifically suitable for publication and will be formally accepted for publication once it meets all outstanding technical requirements.

Kind regards,

Simon J Clark, D.Phil.

Academic Editor

PLOS ONE

Reviewers' comments:

Reviewer's Responses to Questions

**Comments to the Author**

1. If the authors have adequately addressed your comments raised in a previous round of review and you feel that this manuscript is now acceptable for publication, you may indicate that here to bypass the “Comments to the Author” section, enter your conflict of interest statement in the “Confidential to Editor” section, and submit your "Accept" recommendation.

Reviewer #1: All comments have been addressed

2. Is the manuscript technically sound, and do the data support the conclusions?

Reviewer #1: Yes

3. Has the statistical analysis been performed appropriately and rigorously? 

Reviewer #1: Yes

4. Have the authors made all data underlying the findings in their manuscript fully available?

Reviewer #1: Yes

5. Is the manuscript presented in an intelligible fashion and written in standard English?

Reviewer #1: Yes

6. Review Comments to the Author

Reviewer #1: Thank you for your thorough and obviously hard work on making these amendments to the manuscript.

I think it is now much clearer as the the study aims and design, and the data are clearer for the reader (or at least me) to understand and interpret.

This is an interesting study, and is now ready for publication in PlosONE.

7. PLOS authors have the option to publish the peer review history of their article (what does this mean?). If published, this will include your full peer review and any attached files.

Reviewer #1: No

---

## [Editor Report · Acceptance letter]

24 Jul 2020

PONE-D-20-05117R1 

Plasma Biomarkers of the Amyloid Pathway Are Associated with Geographic Atrophy Secondary to Age-Related Macular Degeneration 

Dear Dr. Lashkari:

I'm pleased to inform you that your manuscript has been deemed suitable for publication in PLOS ONE. Congratulations! Your manuscript is now with our production department. 

Kind regards, 

on behalf of

Prof. Simon J Clark 

Academic Editor

PLOS ONE